# Rock Steady Boxing: A qualitative evaluation of a community exercise program for people with Parkinson's disease

Elizabeth W. Regan[1]*, Olivia Burnitz[2], Jessica Hightower[3], Lauren Dobner[4], Alicia Flach[1]

1 Department of Exercise Science, University of South Carolina, Columbia, South Carolina, United States of America, 2 Prisma Health, Columbia, South Carolina, United States of America, 3 ProMotion Rehab and Sports Medicine, Columbia, South Carolina, United States of America, 4 Carolinas Rehabilitation, Concord, North Carolina, United States of America

☯ These authors contributed equally to this work.
* eregan@mailbox.sc.edu

## Abstract

### Purpose

Regular exercise can reduce the symptoms of Parkinson's disease, a common neurodegenerative disorder. The Rock Steady Boxing organization created an exercise program for people with Parkinson's Disease (pwPD) modelled after traditional boxing. The purpose of this study was to better understand the physical function, exercise beliefs, contextual factors, class experiences and outcome perceptions of pwPD participating in Rock Steady Boxing.

### Materials and methods

A pragmatic qualitative approach of semi-structured interviews and class observations was supplemented by cross-sectional physical function measures.

### Results

Ten study participants were community dwelling adults with Parkinson's Disease, participating in Rock Steady Boxing two to three times a week for an average of 10.6 (6.2) months. Most participants (7/10) had good mobility with Timed Up and Go scores below the 14.8 second average for community dwelling pwPD (7/10) and Parkinson's Disease Questionnaire-39 Mobility Scores < 50% (8/10). Participants (9/10) had good exercise self-efficacy, with average scores on the Short Self-efficacy for Exercise Scale > 3/5. Thematic analysis revealed a history of exercise and strong exercise beliefs positively influenced participation. Rock Steady Boxing provided valued social interactions and offered individualized attention through personalized modifications.

**Data Availability Statement:** All data are available at Open Science Framework: https://osf.io/g9j34/?

view_only=d14369481b47408ba4e8fe7cd3355 ? 353.

**Funding:** This work was supported by the University of South Carolina Behavioral-Biomedical Interface Program (National Institute of General Medical Sciences/National Institutes of Health T32 2T326M081740-11A1), 2019 American Heart Association Pre-Doctoral Fellowship.

**Competing interests:** The authors have declared that no competing interests exist.

## Conclusions

Instructor enthusiasm, program modifiability and social support drive participation and provide a challenge for differing abilities. RSB is a valued community offering for pwPD.

## Introduction

Parkinson's disease (PD) is the second most common neurodegenerative disorder in the world, involving progressive motor and non-motor symptoms [1, 2]. While medication is used to manage symptoms of PD, exercise is currently the only intervention with evidence to slow disease progression [3–5]. In addition, regular exercise has been shown to improve gait, balance, strength, flexibility, cardiovascular fitness, and quality of life for people with PD (pwPD) [6–8]. Social interaction, personal fulfilment and learning adaptations related to exercise can also contribute to positive neurologic adaptations and delays in functional decline [9–11].

Due to the degenerative nature of PD, pwPD tend to gradually become less active [12]. Up to 83% of pwPD do not meet the 150-minute weekly physical activity (PA) recommendation [6, 13, 14]. This contrasts with the general U.S. population, of which 47% do not meet the general guidelines [15]. Over time, pwPD do not sustain their baseline levels of activity, suggesting that an increase in disease severity results in a decline in PA [7, 16, 17].

Several factors greatly influence exercise behavior and activity adherence. Barriers include poor physical health, lack of time, fear of falling, low outcome expectations, and low exercise self-efficacy [18–22]. Group programs can offer social support and enjoyment to potentially improve adherence to exercise behaviors and improve outcomes expectations [19, 20, 23, 24].

Rock Steady Boxing (RSB), based in Indianapolis, IN, was the first program in the U.S. for pwPD modelled after traditional boxing training [25]. RSB is unique because was designed specifically for pwPD. Approximately 43,500 people participate in RSB across 900 locations globally [26].

RSB instructors complete a three-day certification program before offering classes at community fitness facilities [25]. All class participants must have a PD diagnosis and a physician referral to participate [25]. RSB classes target agility, speed, muscular endurance, accuracy, hand-eye coordination, functional mobility, footwork, and overall strength [25]. Existing studies demonstrate boxing activities may improve in physical function, balance and quality of life (QOL) for pwPD [27–29].

The purpose of this study was to better understand the physical function, beliefs, and perceptions of contextual factors, class experiences, and outcomes of pwPD while participating in RSB. Study aims were to (1) understand participant physical function (balance, mobility), (2) understand contextual and environmental factors influencing participation, (3) evaluate perception of RSB experience, and (4) assess how pwPD perceived program outcomes. Results can provide insight for recruitment and participant outreach, and influence future community exercise programs for pwPD.

## Materials and methods

### Study design

The study used a pragmatic qualitative approach supplemented with cross-sectional participant surveys and physical function measures (Table 1). Single-session quantitative measures, participant questionnaires, and demographic data were collected to describe the sample

**Table 1. Research aims and study questions.**

| Aims | 1. To understand physical function of RSB participants | 2. To understand contextual factors impacting participation and adherence to the RSB program. | 3. To evaluate participant perception of experience in RSB | 4. To evaluate perceived outcomes of the RSB program |
|---|---|---|---|---|
| Study Questions | A. What are participant's balance level and fall risk?<br>B. What is their functional mobility level?<br>C. How do objective measures compare to self-report measures and stated perceptions | A. What are the barriers to participation in RSB and how do the participants overcome them to regularly participate?<br>B. What are the motivators for participation?<br>D. What impact does caregiver and family support have on participation in the program?<br>E. How influential are environmental factors on participation in RSB?<br>F. How does exercise experience, beliefs and self-efficacy influence participation? | A. What are the activity components of the RSB program? Are there particular components participants find most beneficial/impactful?<br>B. What do participants enjoy or not enjoy about the program?<br>C. Do they feel safe during the program? Do they feel the effort level required is appropriate?<br>D. What are the relationship and social components of RSB? | A. What are the perceived outcomes related to mobility, health, and wellbeing?<br>B. Do participants believe it makes a difference in their symptoms and disease progression? |
| Outcome Measures | • Timed Up and Go<br>• Timed Up and Go Cognitive<br>• Fullerton Advanced Balance Score<br>• PDQ-39 Mobility Sub-score<br>• Qualitative interviews | • PDQ-39 Social Support Sub-score<br>• Short Self-Efficacy for Exercise Scale<br>• Short Outcome Expectations for Exercise Scale<br>• Qualitative interviews | • Qualitative interviews | • Qualitative interviews |

Rock Steady Boxing (RSB), Parkinson's Disease Questionnaire 39 (PDQ-39).

population and contextualize the qualitative data. Semi-structured interviews were the qualitative data collection tool, aided by structured RSB class observations (S1 Table in S1 File). The University of South Carolina Institutional Review Board approval was obtained prior to data collection. Written informed consent was obtained from all participants. All data have been made available at Open Science Framework [30].

## Program description

A RSB program at a community fitness facility in a metropolitan area of South Carolina was the site of study participant recruitment. Classes were offered by certified instructors five times a week with options for different intensity levels. The cost for RSB class participation was a one-time $50.00 fee for the initial orientation session and then $98 a month for up to 20 classes per month.

## Sampling and recruitment

Participants were recruited through criterion sampling from May 4 through June 30, 2019. Researchers attended multiple class times to offer participation to all members. Flyers were provided and included a description and goals of the study, Voluntary participation, the ability to withdraw at any time, and confidentiality were emphasized. Inclusion criteria were to have attended RSB classes at least once a week for at least three months and communicate in English. Participants were excluded if they could not travel to the research lab for the assessment and interview. Participants completed a written informed consent before beginning data collection at the research lab, witnessed by one researcher (OB, JF or LS).

## Quantitative measures and data analysis

To gain more insight into participants' overall quality of life, exercise beliefs and impairment level, this study included cross-sectional measures collected at a campus research lab.

**Patient reported measures.** Participants completed several questionnaires using Research Electronic Data Capture (REDCap version 9.3.6—© 2019 Vanderbilt University):

1. Intake form (S1 Fig in S1 File): basic demographic information, medical history, symptom history.

2. PDQ-39 Mobility and Social Support Subscales: The PDQ-39 is widely used to assess quality of life in pwPD [31]. The PDQ-39 consists of eight subscales, mobility and social support were used. Dimension scores are percentages. Higher scores indicate more impairment [31, 32].

3. The Short Self-Efficacy for Exercise (SSEE) Scale and the Short Outcome Expectations for Exercise Scale (SOEE): The SSEE and SOEE assess confidence in performing exercise behaviors and beliefs about the positive outcomes of exercise [33]. Scores are averaged and can range from 1 (no confidence/no belief in importance of exercise) to 5 (high confidence/ high belief in importance of exercise).

**Functional measures.**

1. Timed up and go (TUG): The TUG is a common clinical assessment used to assess mobility in people with disabilities, including PD [34]. A faster time indicates higher function. Participants started in a seated position. The time it took to rise from a seated position, walk three meters, turn and return to sitting in the chair was recorded [34].

2. TUG cognitive: The TUG was repeated with an additional cognitive task of counting backwards by threes [35].

3. Fullerton advanced balance (FAB) scale: The FAB is a standardized assessment of balance in older adults [36]. Scores can range from 0–40 and higher scores indicate higher function.

**Data analysis.** Mean and ranges were calculated for demographic and RSB attendance data. Measures were compared to normative values where available and counts of participants above and below normative values were reported.

**Qualitative measures and data analysis.** The research team consisted of three female Doctor of Physical Therapy students (OB, JF and LS), a female qualitative mentor who is a physical therapist with a Doctor of Physical Therapy degree and a PhD with qualitative training and experience (ER), and a female subject matter mentor who is a physical therapist with a Doctor of Physical Therapy Degree and is a Board-Certified Specialist in Neurologic Physical Therapy (AF). Mentors trained student researchers in qualitative methods and collaborated on data analysis.

During the study period, researchers (OB, JF and LS) completed structured observations during a RSB session and documented findings on standard form (S2 Fig in S1 File). Observations included describing the class structure, participant interactions and safety. Researchers did not know participants prior to class observation.

Qualitative data collection was an in-depth interview using a semi-structured interview guide (S1 Table in S1 File) by researchers (OB, JF and LS). Audio recordings of semi-structured interviews were de-identified and transcribed verbatim by researchers (OB, JF and LS), and input into a qualitative software program (NVivo version 12 Plus, QSR International,

Melbourne, Australia). All data were identified by assigned participant numbers only. Additionally, structured observations were input into the software. Data analysis, including collaboration and peer review, was completed with guidance from a mentor (ER and AF). Thematic analysis utilized deductive categorical analysis with open coding and inductive analysis within those categories [37], and followed these steps [38]:

1. Initial codes were created for each participant close to the participant's original wording and categorized as a contextual factor, experience, outcome, or other. Initially, one transcript was completed as a group (OB, JF and LS). Coding was reviewed with faculty mentor (ER) to evaluate codes and establish a coding style. The remaining initial coding was completed individually (OB, JF and LS).

2. Researchers (OB, JF, and LS) collaborated with faculty mentors (ER and AF) to group common ideas into themes in a hierarchical manner, including divergent themes when found. Themes were reviewed and revised iteratively.

3. Researchers (OB, JF, and LS) compared resultant themes to each participant's quantitative scores to look for associations.

### Theory

The World Health Organizations International Classification of Function (WHO-ICF) and the social cognitive theory were the theoretical basis for the research questions, interview guide, and the organization of results [39–42]. The WHO-ICF classifies individual's body functions, activities (abilities and limitations), participation (abilities and limitations), environmental factors and personal factors within the context of disease diagnosis [39, 42]. Social cognitive theory evaluates how an individual's self-efficacy, motivation, barriers, and facilitators interact to impact health behaviors [40, 41]. Health behavior participation for people with disability has the additional factor of meaningfulness or reasons to improve their health [39]. This sense of meaning is tied to self-efficacy in managing disability and access to participation opportunities [39].

## Results

### Rock Steady Boxing class description

Structured observations revealed class sizes of 5–13 participants in addition to care partner assistants. Activities included a dynamic warm up, exercise, and a cool down. The exercise portion included five stations of functional activities and boxing activities each with three-minute exercise and one-minute rest intervals. Additional observation noted activities included participant encouraged high volume vocalizations and exercise combined with a cognitive task.

### Study participants

Of the approximately 25–30 regular class participants, ten individuals (8 males, 2 females) participated in this study. All completed an informed consent form. Average participant age was 71.3 years old (range 60–80 years old). Participants attended RSB on average 2–3 times per week and the average length of overall RSB attendance was 10.6 months (range 4–24 months).

### Cross-sectional results

The majority (7/10) of the participants had good mobility with TUG scores faster than the 14.8-second average for community-dwelling pwPD [43] and PDQ-39 Mobility Dimension

**Table 2. Participant demographics.**

| Pt | Age (years) | Sex | Work Status | Length attending RSB (months) | Frequency of attendance (days/ week) | PDQ-39 Mobility Sub-score (%) | PDQ-39 Social Support Sub-score (%) | Exercise confidence (SSEE) (1–5) | Exercise beliefs (SOEE) (1–5) | TUG (sec) | TUG-C (sec) | FAB (0–40) |
|----|------|-----|------|------|------|------|------|------|------|------|------|------|
| 1 | 73 | M | R | 24 | 2 | 7.5 | 0.0 | 4.0 | 4.6 | 8.52 | 9.39 | 32 |
| 2 | 76 | M | R | 18 | 2 | 57.5 | 0.0 | 4.0 | 5.0 | 126.5 | Unable to complete | 7 |
| 3 | 80 | M | R | 7 | 2 | 0.0 | 8.3 | 5.0 | 5.0 | 15.0 | 18.50 | 28 |
| 4 | 68 | M | R | 4 | 2 | 30.0 | 0.0 | 4.0 | 4.8 | 8.19 | 8.75 | 32 |
| 5 | 78 | M | R | 12 | 2 | 7.5 | 0.0 | 3.6 | 3.4 | 11.36 | 17.85 | 27 |
| 6 | 60 | M | R | 6 | 2 | 2.5 | 0.0 | 4.8 | 4.8 | 8.19 | 13.46 | 33 |
| 7 | 74 | M | R | 8 | 3 | 45.0 | 66.7 | 4.0 | 5.0 | 11.53 | 13.55 | 30 |
| 8 | 74 | F | R | 8 | 2–3 | 72.5 | 0.0 | 2.4 | 3.8 | 35.2 | 40.82 | 6 |
| 9 | 60 | F | FT | 12 | 2 | 0.0 | 0.0 | 5.0 | 5.0 | 7.85 | 9.68 | 38 |
| 10 | 70 | M | R | 7 | 2 | 5.0 | 0.0 | 4.8 | 3.6 | 10.11 | 20.14 | 36 |

Data presented as age (years), biological sex (M = male, F = female), work status (R = retired, FT = working full time), Parkinson's Disease Questionnaire-39 (PDQ-39) with lower scores indicating better quality of life, Short Self-Efficacy for Exercise (SSEE) with 5 = greatest confidence, 1 = least confident), Short Outcomes for Exercise Self-Efficacy (SOEE) with 5 = high belief in positive outcomes of exercise, 1 = low belief in the positive outcomes of exercise, timed up and go (TUG) and timed up and go-cognitive (TUG-C) with lower times indicating better mobility, seconds (sec), and the Fullerton Advanced Balance Scale (FAB) with higher scores indicating better balance.

Scores < 50% [32]. Most (8/10) had reduced risk for falls with FAB balance scores > = 27 seconds [36] and many (6/10) below the suggested cutoff for reduced risk of falls on the TUG (<11.5 seconds) [44]. The majority of participants (9/10) had high exercise self-efficacy (SSEE), with only one participant's (participant 8) confidence lower than 3/5. All participants had some belief in the positive benefits of exercise (SOEE), with all scores greater than 3/5. The majority rated their social support high on the PDQ-39 Social support dimension, with 8/10 scoring 0% and one scoring 8%. Only one participant reported low social support with a score of 67%. Study participant demographics, clinical tests and self-report measures are provided in Table 2.

## Qualitative results

Participants completed a single semi-structured interview in a research lab environment privately, with interviews lasting 30–40 minutes. Participant 2's care partner was present due to some verbal communication limitations. Themes are organized around study aims and highlight contextual factors, participant experience, and perceived outcomes. The resulting conceptual framework is presented in Fig 1. Additional participant quotes for each theme are presented in the S2 Table in S1 File. Data saturation was not formally assessed as maximum voluntary participation was achieved. Researchers believe saturation was achieved based on suggested numbers for a homogeneous population and in-depth themes [45].

**Environmental contextual factors.** Environmental factors impacting the RSB experience are all facilitators and include (1) hearing about RSB from a trusted source, (2) RSB as a priority in participant's schedules, (3) family and care partner support, and (4) transportation available.

1. Hearing about RSB from a trusted source: Participants were evenly split between referral to RSB by a healthcare provider or through word of mouth. Those who heard from a healthcare provider cited nurses, hospital staff or physical therapists as the primary source, while

**Fig 1. Conceptual framework for Rock Steady Boxing participation in people with Parkinson's disease.** PD = Parkinson's Disease, RSB = Rock Steady Boxing.

those who heard through word of mouth heard either from a friend or while attending a support group for pwPD.

2. RSB as a priority in participant's schedules: For all ten participants, RSB was a priority; they made time in their schedules to avoid conflict with other activities.

**Participant 8:** "Well, occasionally something comes up, but we've pretty much blocked out those time frames."

3. Family and care partner support: Care partners and family provided encouragement to participants to exercise and to work on maintaining and improving their mobility. Nine of the ten participants reported their spouse was their primary care partner.

**Participant 4:** "she [my wife] gets on me about not—when I'm not exercising sometimes and I need to. . .she supports. . .encouraged me to start it to start with. . .she gets me here. . .and I can tell that she wants me to improve. . .or maintain. . .mainly what I'm concerned about is maintaining and. . .not regressing. . .She is very supportive, gets after me when I'm not—when I'm laying around too much."

**Participant 6:** "I was bitter. . .I was angry. . .and then, with the help of my wife and my family and all that, and my friends, I snapped out of it. . .I'm where I am today, and then I joined Rock Steady."

4. Transportation available: Access to transportation to RSB supported attendance. Half of the participants drove themselves, and half relied on their spouse or family members.

**Participant 10:** "I drive. My wife comes along most of the time. But I drive. . .Not that I don't want her support, it's just that I want to try to do it as long as I can."

**Personal contextual factors.**   Personal factor themes include (1) positive beliefs and history of exercise as motivators, (2) focused foals as motivators, and (3) RSB participation despite frustrations with PD symptoms, functional limitations, and disease progression.

1. Positive exercise beliefs and history of exercise as motivators: Seven of the ten participants reported a long history of being physically active. However, nine participants had only exercised individually prior; RSB was their first group exercise experience. Three participants said they currently perform other physical activity in addition to RSB. Generally, participants believed that exercise was important or vital for maintaining their function while living with PD. In contrast, one participant said they were not interested in exercise before starting RSB.

**Participant 8:** "I knew it was beneficial but in terms of overall, it's good for everybody. But I didn't realize or understand the specific benefits that it is for patients by virtue of as my if I understand it correctly, 20–30 minutes of strenuous exercise helps with reduction of dopamine, which is good."

**Participant 9:** "Well I've always been very active. I've always wanted to do something. I've always wanted to stay fit. I've always wanted to be strong."

2. Focused goals as motivators: Various goals were found to contribute to participation in RSB including to improve function and/or continue participation in activities (six participants) and to maintain and/or slow disease progression (four participants).

**Participant 1:** "To feel better, ya know, get more exercise in."

**Participant 2:** "Get my golf game back."

**Participant 5:** "My goal is to never ever use a cane, that's my goal."

3. RSB participation despite frustrations with PD symptoms, functional limitations, and disease progression: Many participants were frustrated due to experiencing limitations. These include being down or depressed, and new limits in functional activities and recreational participation. Despite these barriers, participants continued to pursue and participate in RSB. Three reported that the diagnosis caused depression.

**Participant 6:** "I was bitter. I was angry. I'm embarrassed to say that I was questioning my faith. Um. . .that bothered me a lot. I would go for two, two to three months straight I would go from the bedroom to my lounge chair. And I was having medication issues. . . So, I don't know if it was a combination of all that and then the diagnosis and the bitterness, but I, I was, a vegetable for—for three or four months."
Many participants discussed how the progression of their disease limits their functional activities. Deficits reported included balance (three participants), communication (two participants), strength and flexibility (three participants), and tremors (two participants).

**Participant 1:** [In regard to moving from sitting to standing] "That's probably the worst thing right there. That has, that has kind of gotten a little worse."

**Participant 5:** "Balance and stuff, I'm not very good. And the other thing, I can do it, but I'm not real strong at getting up off my knees. I have to push up, and then sometimes I get a little push there. The other day I fell doing it."
When asked about recreational activities and work, six said they were either forced to stop or decrease their desired recreational activities due to PD. Seven participants retired before they were diagnosed and one retired because of PD.

**Participant 6:** "I just couldn't do my, my job anymore. And the stress of it. It was a very stressful job, and that made my tremors, Parkinson's worse. . .they allow me to retire, medically retire they called it."

**Participant 10:** "I used to bowl. Yeah. And now, ya know, the balance with the bowling is, is a challenge."

**Participant experience.** In class experience themes highlighted the (1) varied class attributes and ability to modify contributes to a positive experience, (2) care partner presence as a necessary component of class, (3) varied physical and mental responses during class, (4) relationships during class facilitate positive experiences and regular attendance and (5) positive results contribute to regular attendance.

1. Varied class attributes and ability to modify contributes to a positive experience: Eight of the ten participants reported that they felt confident that with the help of the instructors, they would be able to modify the activities as needed according to their specific abilities or limitations.

**Participant 2:** "They have modifications. . .for different levels. So [other participant] will sit on something, where other people are standing"

**Participant 3:** "I modify so it's a little harder"
Participants also reported enjoying the variety in class components.

**Participant 9:** "There's some simple little things, simple little things. There's like this one thing that has like little screws, and you put the little washers on it, and it seems stupid. But you know what, as fast as you can do it, get your little fingers moving and you do all these little things. And we had to do all these little things. It just seems silly, but those are things that people do like at a picnic or people might play here."

2. Care partner presence as a necessary component of class: Three participants reported that care partner presence was a large facilitator for successful participation in the class, which was confirmed by structured observations. Structural observations also noted that care partner presence facilitated safety.

**Participant 4:** "The wives will. . .be at each station, and um. . .show you what you're supposed to do at that station if you don't remember or. . . they go over it before the class starts, still it's hard to remember which. . . Hard to remember without having to look. . .look at the board every time."

**Structured Observation:** "Care partners that were present were very interactive with all participants, encouraging and assisting others besides the person they arrived with. They also assisted the instructors with setup and breakdown of materials."

3. Varied physical and mental responses during class: Participants reported a wide variety mental and physical responses during class, including feeling mentally clear and enjoying the class (four participants), frustration or stress during the class (three participants), fatigue during class (four participants) or increased energy/strength during class (three participants). All 10 participants reported feeling safe during class.

**Participant 7:** "I feel like I have a lot of energy [during class]."

**Participant 8:** "About halfway through, I get real tired and the second half is kind of, is a real push. . .But it's due to fatigue in my legs, weakness in my legs. . .that I and I just have to keep pushing or do some seated activities for a few minutes."

4. Relationships during class facilitate positive experiences and regular attendance: Interactions were frequently reported as a contributor to a positive experience, including participant-participant (all participants), participant-instructor (all participants) and participant-care partner interactions (three participants). Many referenced the enthusiasm of the instructors, as well as the ability of the instructors to adapt components of the class to meet individual needs.

**Participant 4:** "The wives, a lot of them help out. . . if they didn't have the wives there, might not have enough people."

**Participant 5:** "Everybody is accepting, everybody knows everybody, nobody's judging anybody. And when you come here you feel welcome, whether you want to be here or not. They all, ya know, the, the um, the Parkies, as they call themselves, all speak to each other. They're respectful and kind."

**Participant 8:** "[The instructors] are very vivacious and I'm amazed at the energy level that they maintain. . .and just their excitement and enthusiasm is a positive

reinforcement. . .they're always high-fiving and "good job" and always encouraging, uh, participation in what we've done."

5. Positive results contribute to regular attendance: Eight participants stated that one of the reasons they continue to attend RSB is that they see positive results from the class, which is discussed in detail in the outcomes section.

**Perceived outcomes.**  Participants described various outcomes they experienced following their participation in RSB. Some noted improvements, while others experienced no change. Themes include (1) affirmation or improvement in exercise self-efficacy, (2) functional improvements in gait and balance, but other physical and mobility symptoms remain, and (3) fatigue or increased energy after class.

1. Affirmation or improvements in exercise self-efficacy: Six participants expressed affirmed or improved beliefs regarding the benefits of exercise, especially for pwPD. Five participants reported an increase in exercise self-efficacy. Participants found it helpful to see others with the same disease process exercising.

   **Participant 5**: "It's affirmed, and it's made me exercise more, maybe a little more than I would have."

   **Participant 6**: "This is a, this is a strong class, and um, uh, I feel like I've accomplished something."

2. Functional improvements in gait and balance, but other physical and mobility symptoms remain: All participants described improvements in function, such as activities of daily living, balance, gait, and strength. Gait was the most reported functional improvement (seven participants).

   **Participant 1:** "I haven't fallen as much. I used to fall every, I used to fall five or six times a week. Now I fall maybe once every two weeks."

   **Participant 8**: "It gives me a little more steadiness, a little more strength and endurance." Although all participants reported functional gains from RSB, the majority (eight participants) also discussed remaining physical impairments.

   **Participant 10: "**I experience tremors more now as it's going along. I used to take three pills a day and now I'm taking 5. Ya know, it's just that I need them."

3. Fatigue or increased energy after class: Most participants reported experiencing fatigue following RSB. Only three reported a quick recovery with minimal to no fatigue. However, several require some rest or a nap before going on with their day.

   **Participant 4**: "After my nap, at the house. . .spend 2 hours. . .rest and. . .get back to doin' things."
   Four participants, however, had the opposite response, reporting increased energy during or following class.

**Participant 9**: "I always feel good when I leave here. I really do. I always feel good when I leave here."

## Differences between functional measures, patient-reported outcomes and qualitative results

For some participants, the findings on the measured questionnaires and physical outcome measures contrasted perceptions reported in the interview. Participant 8, who had low exercise self-efficacy scores (2.25/5), a slow TUG indicative of mobility and strength difficulties (35.2 seconds), and low FAB reflecting poor balance (6/40), reported positive experiences and improvements in her strength and balance with the program during her interview. Participant 7 scored the lowest on social support on the PDQ-39 (66.67%) and then discussed a supportive care partner who encourages exercise and drives him to the program during the interview. Participant 1 had a fast TUG (8.52 seconds), which includes sitting to standing, but reported difficulty with sit-to-stand movements in the interview. Varying findings were present in two of the three outcome themes with physical improvements opposing remaining symptoms, and post-class fatigue contrasting improved energy. Both participant 3 and 6 noticed some worsening of PD symptoms but wondered in their interviews if the decline would be more severe without RSB participation.

**Participant 6**: ". . .uh, has the class made a difference, would I have gotten to the point where I couldn't do something, and the class has stopped the progression of that? It's hard —I don't know. I can't say."

## Discussion

The results of this study indicate that regular participation in RSB is influenced by a history of exercise, the ability to overcome barriers, enjoyment of varied, challenging modifiable activities, care partner support, and social interaction. These findings can influence clinician behavior to maximize attendance at RSB and similar programs and improve the quality of life for pwPD.

### Exercise experience and belief in benefit as key motivators

Consistent with existing literature, a history of regular exercise prior to PD diagnosis was a key contributor to participation in RSB for study participants as was a belief that exercise was important to manage their PD symptoms [46, 47]. Most reported a long history of being physically active, though only one had participated in group exercise before joining RSB. Group exercise such as RSB may be especially beneficial for pwPD that have been forced to stop or decrease their participation in other recreational activities [19, 20, 46]. Exercise as the best method for slowing disease progression was common knowledge for study participants, emphasizing the importance of education on exercise benefits by clinicians treating pwPD [46, 48]. The need for this education is especially important for those who do not have history of exercise and an existing belief in its value before their diagnosis [46, 48].

### Overcoming barriers and optimizing facilitators

Participants either did not face common barriers (transportation, cost, time, lack of support) or were able to avoid other common barriers (scheduling, PD symptoms) [46, 49]. Cost was so

rarely mentioned as a barrier or facilitator, it did not rise to a theme level. For this group of regular participants, the lack of these common barriers to exercise participation likely facilitated their on-going participation. In addition to transportation, care partners also provided encouragement and in-class assistance. This combination of factors illustrates that the participants in our study greatly value their time at RSB and intentionally prioritize participation.

## RSB may fill a void of lost activities

Feelings of frustration, depression, and powerlessness are common after being diagnosed with PD and are not associated with disease severity or progression [50, 51]. In the current study, six participants stated signs and symptoms of PD they experienced forced them to stop or reduce at least one activity that they previously enjoyed. Participation in RSB may provide an improved sense of control for pwPD by engaging in a new activity that is both challenging and empowering. For example, participant 5 described being unable to build model ships due to experiencing tremors associated with PD but reports his hand-eye coordination has improved since starting RSB. Additionally, participant 6 described feeling "bitter" and "angry" after being diagnosed with PD, but that RSB helped him "snap out of it." These reports indicate that RSB may fill a void left by the loss of activities and serve to boost their exercise self-efficacy and quality of life.

## Unique characteristics of RSB program drives participation

The high-intensity varied activities that are a hallmark of RSB may contribute to its success [52]. Additionally, enjoyment, social support, adaptability, and supervision are factors in participation [19, 20]. All participants spoke about the importance of the relationships in class, including participant-participant, participant-care partner, and participant-instructor interactions. Social support is especially important in addressing the isolation, depression and apathy that can occur for pwPD [50, 51]. The appreciation of the group dynamic is also congruent with the study by Faherty et al., suggesting that an "enriched environment", which incorporates social interactions into exercise, may be even more beneficial for pwPD than just exercise alone [10]. PwPD who desire greater social interaction or struggle with social isolation or apathy may achieve greater benefit from a program like RSB than individual exercise [40].

## Perception of physical abilities differed from functional measures

An unexpected finding in the current study was the differences in perception of function and measured outcomes. Quantitative assessment is still the dominant method in most healthcare settings [53], but the present study suggests that healthcare providers should view functional status measures in context of self-report of status, recent changes and future goals. Incorporating the individual's perspective provides rich insights into the true experience of PD, and ensures education and treatment can be individually targeted for the highest possible quality of life for pwPD.

Knowledge of individual activity level expectations is another key piece of information when evaluating mobility status. Not knowing the activity levels of a pwPD before their diagnosis can impact how a clinician or researcher might evaluate a certain outcome measure score. For example, someone with what is considered a "good" score on the TUG, may find even a small deficit in mobility to be debilitating if they were functioning at very high levels of movement prior. Additionally, the degenerative nature of PD, individual variations, and the potential to slow decline with exercise can be confusing to pwPD who may only notice their increasing deficits. Clinicians can support these discrepancies through regular assessment, sharing results and discussing individual activity and life participation impacts. This person-

centered approach can help maintain or improve self-efficacy and motivation for exercise for pwPD [40, 48]. RSB exercise professionals can assist by understanding goals for participating and completing regular evaluations and re-evaluations of the FAB and the TUG. Regular reassessment with feedback provided to the pwPD can help participants focus on their abilities [48]. In addition, significant reductions in performance in either test could prompt a referral to physical therapy for a new plan of care. In this context, the physical therapist could then provide recommendations for new modifications to the RSB program for the individual.

### Clinicians supporting participation in RSB

The results of this study describe individuals who may be most successful at RSB or a similar program: a pwPD with good social support, a previous history of exercise, positive exercise beliefs, and available transportation. RSB and similar programs may also serve as a less expensive option than frequent visits to healthcare providers or one-on-one personal training. As class attendees perceive RSB providing individualized modifications and attention, RSB is a more affordable option. Still, this cost may remain a barrier for some. Incorporating RSB into community fitness centers with sliding scale income-based memberships may further reduce the financial barrier.

Healthcare providers can assist pwPD by partnering with local RSB programs to provide direct referrals and education for exercise professionals. Referral from a trusted provider to RSB was a facilitator for current study participants [46, 48]. Healthcare providers can also collaborate with exercise professionals to provide education on PD and when to refer a participant to a physician or physical therapist. Intentional collaboration between healthcare providers and community exercise professionals can improve access and quality of group exercise programs and drive patient-centered care [14, 54, 55].

### Study limitations and future research

The current study was limited to a small sample of successful regular participants in one RSB program, limiting generalizability. Additional research on those who desire to participate in RSB but are unable, and following participants over time will provide further knowledge of the influence of RSB on pwPD. Future studies with larger sample sizes that include pre-post measured outcomes with a control group would further establish program outcomes [56]. Additionally, future studies can include qualitative data along with quantitative outcomes to capture the rich experiences and perceptions of pwPD. Mixed methods can build a more complete understanding of the impact of RSB.

### Conclusion

RSB participants value exercise, have exercise self-efficacy, and have support to participate. Despite the frustration and emotional distress many pwPD experience because of the diagnosis and symptoms of PD, participants overcome these barriers to combat disease progression. The RSB program fosters community PA for pwPD through participation in an enjoyable activity, perceived improvements in function and mobility, and positive social interactions. The enthusiasm of the instructors, modifiability and social support incorporated into the RSB program allow participation at differing abilities and provide a challenge. Clinical providers can support pwPD through education and collaboration with RSB programs to facilitate successful long-term exercise and health. Community group exercise programs can use a similar model to RSB to engage other individuals with neurologic degenerative diseases and chronic mobility impairments.

## Supporting information

**S1 File.**
(DOCX)

## Author Contributions

**Conceptualization:** Elizabeth W. Regan, Alicia Flach.

**Data curation:** Elizabeth W. Regan, Olivia Burnitz, Jessica Hightower, Lauren Dobner, Alicia Flach.

**Formal analysis:** Elizabeth W. Regan, Olivia Burnitz, Jessica Hightower, Lauren Dobner, Alicia Flach.

**Methodology:** Elizabeth W. Regan.

**Supervision:** Elizabeth W. Regan, Alicia Flach.

**Writing – original draft:** Elizabeth W. Regan, Olivia Burnitz, Jessica Hightower, Lauren Dobner, Alicia Flach.

**Writing – review & editing:** Elizabeth W. Regan, Alicia Flach.

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
