## [Decision Letter · Decision Letter 0]

18 Jun 2024

PONE-D-24-17855Rock Steady Boxing: A Qualitative Evaluation of a Community Exercise Program for People with Parkinson’s DiseasePLOS ONE

Dear Dr. Regan,

Thank you for submitting your manuscript to PLOS ONE. After careful consideration, we feel that it has merit but does not fully meet PLOS ONE’s publication criteria as it currently stands. Therefore, we invite you to submit a revised version of the manuscript that addresses the points raised during the review process.

Based on the reviewers' suggestions, the paper needs major revision.  The reviewers' comments can be found below.

We look forward to receiving your revised manuscript.

Kind regards,

Tanja Grubić Kezele, Ph.D., M.D.

Academic Editor

PLOS ONE

“This work was supported by the University of South Carolina Behavioral-Biomedical Interface Program (National Institute of General

Medical Sciences/National Institutes of Health T32 2T326M081740-11A1), 2019 American Heart Association Pre-Doctoral Fellowship.”

4. In the online submission form you indicate that your data is not available for proprietary reasons and have provided a contact point for accessing this data. Please note that your current contact point is a co-author on this manuscript. According to our Data Policy, the contact point must not be an author on the manuscript and must be an institutional contact, ideally not an individual. Please revise your data statement to a non-author institutional point of contact, such as a data access or ethics committee, and send this to us via return email. Please also include contact information for the third party organization, and please include the full citation of where the data can be found.

Reviewers' comments:

Reviewer's Responses to Questions

**Comments to the Author**

1. Is the manuscript technically sound, and do the data support the conclusions?

Reviewer #1: Yes

Reviewer #2: Yes

2. Has the statistical analysis been performed appropriately and rigorously? 

Reviewer #1: Yes

Reviewer #2: N/A

3. Have the authors made all data underlying the findings in their manuscript fully available?

Reviewer #1: Yes

Reviewer #2: Yes

4. Is the manuscript presented in an intelligible fashion and written in standard English?

Reviewer #1: Yes

Reviewer #2: Yes

5. Review Comments to the Author

Reviewer #1: This study is about the qualitative evaluation of the Rock Steady Boxing (RSB) community exercise program for Parkinson's disease patients. I have the following suggestions:

1. In this study, semi-structured interviews and course observations were used, which are commonly used methods in qualitative research. Suggest the author to consider whether it is necessary to add quantitative data to support qualitative findings.

2. The study involved 10 participants, which is a relatively small sample size. Suggest the author to discuss the potential impact of sample size on research results and consider how to make the results more universal.

3. The results section should clearly present the research findings and compare them with existing literature and other studies. Suggest the author to consider adding comparative analysis to highlight the uniqueness and potential advantages of the RSB project.

5. The discussion section should delve into the meaning of research findings and how they relate to existing theories and practices. It is suggested that the author consider proposing specific practical suggestions and how to translate the research results into strategies to improve the quality of life of Parkinson's disease patients.

6. It is suggested that the author consider whether RSB can serve as an adjunct to the treatment of Parkinson's disease and discuss its potential applications in a wider community.

7. The references are too outdated.

Reviewer #2: This study provides valuable insights into the impact of the Rock Steady Boxing program on individuals with Parkinson’s Disease. Despite its strengths, including a well-defined qualitative approach and practical implications, the manuscript could be improved by addressing its limitations more thoroughly, integrating the qualitative and quantitative data more effectively, and providing clearer directions for future research.

Abstract

The abstract could benefit from a brief mention of the sample size and the duration of the study.

It should include the primary outcomes measured to give readers a quick insight into the key findings.

Introduction

The introduction is lengthy and could be more concise. Some background information could be summarized to maintain focus.

It would be helpful to include more recent references to ensure the literature review is up-to-date, such as:

Bevilacqua R, et al. Rehabilitation of older people with Parkinson's disease: an innovative protocol for RCT study to evaluate the potential of robotic-based technologies. BMC Neurol. 2020 May 13;20(1):186. doi: 10.1186/s12883-020-01759-4

Maranesi E, et al. The Effect of Non-Immersive Virtual Reality Exergames versus Traditional Physiotherapy in Parkinson's Disease Older Patients: Preliminary Results from a Randomized-Controlled Trial. Int J Environ Res Public Health. 2022 Nov 10;19(22):14818. doi: 10.3390/ijerph192214818

Mao Q, et al. Effectiveness of sensor-based interventions in improving gait and balance performance in older adults: systematic review and meta-analysis of randomized controlled trials. J Neuroeng Rehabil. 2024 May 28;21(1):85. doi: 10.1186/s12984-024-01375-0.

Material and methods

The sample size of ten participants is relatively small, which could limit the generalizability of the findings. The authors should acknowledge this limitation more explicitly.

The methodology section could benefit from a more detailed description of the interview and observation protocols to enhance reproducibility.

Results

The results section could be improved by including more direct quotes from participants to enrich the qualitative data.

There is a need for a clearer connection between the qualitative themes and the quantitative measures to provide a more integrated understanding of the findings.

Discussion

The discussion could be more focused, with less repetition of the results.

Potential biases and limitations of the study, such as the small sample size and the subjective nature of qualitative analysis, should be discussed in more detail.

Future research directions should be outlined more explicitly, providing a clear path for subsequent studies.

Additional Comments

Formatting and Style: The manuscript is well-structured and written in a clear. However, some sections could benefit from minor grammatical revisions to enhance readability.

Ethical Considerations: The ethical considerations are well addressed, but more detail on how confidentiality and anonymity were maintained during the qualitative data collection would be beneficial.

Data Availability Statement: The data availability statement needs to be more precise about what data can be shared and under what conditions, ensuring compliance with PLOS ONE’s data policy

6. PLOS authors have the option to publish the peer review history of their article (what does this mean?). If published, this will include your full peer review and any attached files.

Reviewer #1: No

Reviewer #2: No

---

## [Author Response · Author response to Decision Letter 0]

1 Jul 2024

Comments to the Author

5. Review Comments to the Author

Reviewer #1: This study is about the qualitative evaluation of the Rock Steady Boxing (RSB) community exercise program for Parkinson's disease patients. I have the following suggestions:

1. In this study, semi-structured interviews and course observations were used, which are commonly used methods in qualitative research. Suggest the author to consider whether it is necessary to add quantitative data to support qualitative findings.

Author Response: We appreciate the value placed by the reviewer on the qualitative data alone. We feel that the quantitative data helps describe the sample well due to the heterogenous presentation of people with Parkinson’s disease and that it helps contextualize the qualitative findings.

2. The study involved 10 participants, which is a relatively small sample size. Suggest the author to discuss the potential impact of sample size on research results and consider how to make the results more universal.

Author Response: We have expanded the study limitations and future research section to specify the small sample size and the limits in generalizability. 

“The current study was limited to a small sample of successful regular participants in one RSB program limiting generalizability. Additional research on those who desire to participate in RSB but are unable, and following participants over time will provide further knowledge of the influence of RSB on pwPD. Further studies of pre-post measured outcomes with a control group would further establish outcomes of the program.”

3. The results section should clearly present the research findings and compare them with existing literature and other studies. Suggest the author to consider adding comparative analysis to highlight the uniqueness and potential advantages of the RSB project.

Author Response: Authors made changes to the discussion section on “Unique characteristics of the RSB program drives participation” to clarify the unique attributes including the high intensity and variation which has been found to be most effective in addressing Parkinsons, as well as the social characteristics and adaptability. 

“The high intensity varied activities that are a hallmark of RSB may contribute to its success.(52) Additionally, enjoyment, social support, adaptability, and supervision are factors in participation.(20, 21) All participants spoke about the importance of the relationships in class, including participant-participant, participant-care partner, and participant-instructor interactions. Social support is especially important in addressing the isolation, depression and apathy that can occur for pwPD.(50, 51) The appreciation of the group dynamic is also congruent with the study by Faherty et al., suggesting that an “enriched environment”, which incorporates social interactions into exercise, may be even more beneficial for pwPD than just exercise alone.(10) PwPD who desire greater social interaction or struggle with social isolation or apathy may achieve greater benefit from a program like RSB than individual exercise.(41)”

5. The discussion section should delve into the meaning of research findings and how they relate to existing theories and practices. It is suggested that the author consider proposing specific practical suggestions and how to translate the research results into strategies to improve the quality of life of Parkinson's disease patients.

Author Response: Authors appreciate the suggestion and feel that updates to the discussion sections address the concern. Author’s reference social cognitive theory (Schunk and Bandura), and the ICF model (van Uem) as part of the reflection on the satisfaction and success the participants found in their experience with RSB. Authors also discuss in detail how clinicians and exercise professionals leading RSB classes can collaborate to improve the quality of life for people with Parkinson’s disease. 

6. It is suggested that the author consider whether RSB can serve as an adjunct to the treatment of Parkinson's disease and discuss its potential applications in a wider community.

Author Response: Authors address in the “clinicians supporting participation in RSB” section of the discussion how RSB and physical therapy may work together as a patient-centered treatment for PwPD. Authors have also added a sentence to the conclusion on how the RSB model might expand to the greater community. 

“Community group exercise programs can use a similar model to RSB to engage other individuals with neurologic degenerative diseases and chronic mobility impairments.”

7. The references are too outdated.

Author Response:

Author’s have updated clinical and exercise references to more recent citations, replacing references 1 and 2 with the following:

1. Armstrong MJ, Okun MS. Diagnosis and treatment of Parkinson disease: a review. Jama. 2020;323(6):548-60.

2. Bloem BR, Okun MS, Klein C. Parkinson's disease. The Lancet. 2021;397(10291):2284-303.

And the following reference was added related to exercise slowing disease progression and the benefits of exercise:

5. Crotty GF, Schwarzschild MA. Chasing protection in Parkinson’s disease: does exercise reduce risk and progression? Frontiers in aging neuroscience. 2020;12:186.

8. Tsukita K, Sakamaki-Tsukita H, Takahashi R. Long-term effect of regular physical activity and exercise habits in patients with early Parkinson disease. Neurology. 2022;98(8):e859-e71.

11. Alarcón TA, Presti-Silva SM, Simões APT, Ribeiro FM, Pires RGW. Molecular mechanisms underlying the neuroprotection of environmental enrichment in Parkinson’s disease. Neural Regeneration Research. 2023;18(7):1450-6.

The following was added to address enjoyment and improved adherence to group programs:

Chakraverty D, Roheger M, Dresen A, Krohm F, Klingelhöfer J, Ernst M, et al. “There is only one motive… fun.” Perspectives of participants and providers of physical exercise for people with Parkinson’s disease. Disability and Rehabilitation. 2024:1-10

The following replaced older references regarding depression and powerlessness in PwPD:

50. Cong S, Xiang C, Zhang S, Zhang T, Wang H, Cong S. Prevalence and clinical aspects of depression in Parkinson’s disease: a systematic review and meta‑analysis of 129 studies. Neuroscience & Biobehavioral Reviews. 2022;141:104749.

51. Rosa Tdl, Scorza FA. Stigma in Parkinson's disease: Placing it outside the body. Clinics. 2022;77:100008.

Reviewer #2: This study provides valuable insights into the impact of the Rock Steady Boxing program on individuals with Parkinson’s Disease. Despite its strengths, including a well-defined qualitative approach and practical implications, the manuscript could be improved by addressing its limitations more thoroughly, integrating the qualitative and quantitative data more effectively, and providing clearer directions for future research.

Abstract

The abstract could benefit from a brief mention of the sample size and the duration of the study.

It should include the primary outcomes measured to give readers a quick insight into the key findings.

Introduction

The introduction is lengthy and could be more concise. Some background information could be summarized to maintain focus.

Author Response: We have consolidated background information as suggested in the introduction and included the sample size in the abstract. Outcome measure descriptions are included in the results. 

It would be helpful to include more recent references to ensure the literature review is up-to-date, such as:

Bevilacqua R, et al. Rehabilitation of older people with Parkinson's disease: an innovative protocol for RCT study to evaluate the potential of robotic-based technologies. BMC Neurol. 2020 May 13;20(1):186. doi: 10.1186/s12883-020-01759-4

Maranesi E, et al. The Effect of Non-Immersive Virtual Reality Exergames versus Traditional Physiotherapy in Parkinson's Disease Older Patients: Preliminary Results from a Randomized-Controlled Trial. Int J Environ Res Public Health. 2022 Nov 10;19(22):14818. doi: 10.3390/ijerph192214818

Mao Q, et al. Effectiveness of sensor-based interventions in improving gait and balance performance in older adults: systematic review and meta-analysis of randomized controlled trials. J Neuroeng Rehabil. 2024 May 28;21(1):85. doi: 10.1186/s12984-024-01375-0.

Author Response:

Author’s have updated clinical and exercise references to more recent citations, replacing references 1 and 2 with the following:

1. Armstrong MJ, Okun MS. Diagnosis and treatment of Parkinson disease: a review. Jama. 2020;323(6):548-60.

2. Bloem BR, Okun MS, Klein C. Parkinson's disease. The Lancet. 2021;397(10291):2284-303.

And the following reference was added related to exercise slowing disease progression and the benefits of exercise:

5. Crotty GF, Schwarzschild MA. Chasing protection in Parkinson’s disease: does exercise reduce risk and progression? Frontiers in aging neuroscience. 2020;12:186.

8. Tsukita K, Sakamaki-Tsukita H, Takahashi R. Long-term effect of regular physical activity and exercise habits in patients with early Parkinson disease. Neurology. 2022;98(8):e859-e71.

11. Alarcón TA, Presti-Silva SM, Simões APT, Ribeiro FM, Pires RGW. Molecular mechanisms underlying the neuroprotection of environmental enrichment in Parkinson’s disease. Neural Regeneration Research. 2023;18(7):1450-6.

The following was added to address enjoyment and improved adherence to group programs:

Chakraverty D, Roheger M, Dresen A, Krohm F, Klingelhöfer J, Ernst M, et al. “There is only one motive… fun.” Perspectives of participants and providers of physical exercise for people with Parkinson’s disease. Disability and Rehabilitation. 2024:1-10

The following replaced older references regarding depression and powerlessness in PwPD:

50. Cong S, Xiang C, Zhang S, Zhang T, Wang H, Cong S. Prevalence and clinical aspects of depression in Parkinson’s disease: a systematic review and meta‑analysis of 129 studies. Neuroscience & Biobehavioral Reviews. 2022;141:104749.

51. Rosa Tdl, Scorza FA. Stigma in Parkinson's disease: Placing it outside the body. Clinics. 2022;77:100008.

Material and methods

The sample size of ten participants is relatively small, which could limit the generalizability of the findings. The authors should acknowledge this limitation more explicitly.

Author Response: We have expanded the study limitations and future research section to specify the small sample size and the limits in generalizability. 

“The current study was limited to a small sample of successful regular participants in one RSB program limiting generalizability. Additional research on those who desire to participate in RSB but are unable, and following participants over time will provide further knowledge of the influence of RSB on pwPD. Further studies of pre-post measured outcomes with a control group would further establish outcomes of the program.”

The methodology section could benefit from a more detailed description of the interview and observation protocols to enhance reproducibility.

Author Response: Authors have included the interview guide (S2 Table) and the structured observation form (S3 Table) as well as shared the NVIVO file which includes the data in the data sharing statement. 

Results

The results section could be improved by including more direct quotes from participants to enrich the qualitative data.

Author Response: Authors appreciate the recommendation to include more quotes. We have included what we feel are the most important and relevant quotes in the main text, and several more examples for each theme in the supplementary material (S4 Table) should readers wish to dive deeper into the results. 

There is a need for a clearer connection between the qualitative themes and the quantitative measures to provide a more integrated understanding of the findings.

Author Response: Updates were made to the results section noting contrasts to be more explicit, and the potential reasons for these findings is presented in the discussion.

“For some participants, the findings on the measured questionnaires and physical outcome measures contrasted perceptions reported in the interview. Participant 8, who had low exercise self-efficacy scores (2.25/5), a slow TUG indicative of mobility and strength difficulties (35.2 seconds), and low FAB reflecting poor balance (6/40), reported positive experiences and improvements in her strength and balance with the program during her interview. Participant 7 scored the lowest on social support on the PDQ-39 (66.67%) and then discussed a supportive care partner who encourages exercise and drives him to the program during the interview. Participant 1 had a fast TUG (8.52 seconds), which includes sitting to standing, but reported difficulty with sit-to-stand movements in the interview. Varying findings were present in two of the three outcome themes with physical improvements opposing remaining symptoms, and post-class fatigue contrasting improved energy. Both participant 3 and 6 noticed some worsening of PD symptoms but wondered in their interviews if the decline would be more severe without RSB participation.” 

Discussion

The discussion could be more focused, with less repetition of the results.

AUTHORS RESPONSE: Authors have made updates to the discussion to remove repetition and increase focus. 

Potential biases and limitations of the study, such as the small sample size and the subjective nature of qualitative analysis, should be discussed in more detail.

Future research directions should be outlined more explicitly, providing a clear path for subsequent studies.

AUTHORS RESPONSE: The study limitations section was updated to address these concerns, including a reference to a protocol for a large randomized trial of exercise:

“The current study was limited to a small sample of successful regular participants in one RSB program limiting generalizability. Additional research on those who desire to participate in RSB but are unable, and following participants over time will provide further knowledge of the influence of RSB on pwPD. Future studies with larger sample sizes that include pre-post measured outcomes with a control group would further establish outcomes of the program.(54) Additionally, future studies can include qualitative data along with quantitative outcomes to capture the rich experiences and perceptions of pwPD. Mixed methods can build a more complete understanding of the impact of RSB.”

Additional Comments

Formatting and Style: The manuscript is well-structured and written in a clear. However, some sections could benefit from minor grammatical revisions to enhance readability.

AUTHORS RESPONSE: Authors have reviewed the manuscript in entirety and made modifications to enhance clarity where found.

Ethical Considerations: The ethical considerations are well addressed, but more detail on how confidentiality and anonymity were maintained during the qualitative data collection would be beneficial.

AUTHORS RESPONSE: Updated the confidentiality details below: 

In sampling and recruitment:

“Voluntary participation, the ability to withdraw at any time, and confidentiality measures were emphasized. Participants data was de-identified and only a participant number was used.”

In qualitative measures and data analysis:

“Audio recordings of semi-structured interviews were de-identified and transcribed verbatim by researchers (OB, JF and LS), de-identified and input into NVivo 12 Plus qualitative data analysis software program (version 12 Plus, QSR International, Melbourne, Australia). All data was identified by assigned participant numbers only.”

Data Availability Statement: The data availability statement needs to be more precise about what data can be shared and under what conditions, ensuring compliance with PLOS ONE’s data policy.

Author Response: We have made the data fully available in the Op

---

## [Decision Letter · Decision Letter 1]

14 Aug 2024

Rock Steady Boxing: A qualitative evaluation of a community exercise program for people with Parkinson’s disease

PONE-D-24-17855R1

Dear Dr. Regan,

We’re pleased to inform you that your manuscript has been judged scientifically suitable for publication and will be formally accepted for publication once it meets all outstanding technical requirements.

Kind regards,

Tanja Grubić Kezele, Ph.D., M.D.

Academic Editor

PLOS ONE

Additional Editor Comments (optional):

Reviewers' comments:

Reviewer's Responses to Questions

**Comments to the Author**

1. If the authors have adequately addressed your comments raised in a previous round of review and you feel that this manuscript is now acceptable for publication, you may indicate that here to bypass the “Comments to the Author” section, enter your conflict of interest statement in the “Confidential to Editor” section, and submit your "Accept" recommendation.

Reviewer #2: All comments have been addressed

2. Is the manuscript technically sound, and do the data support the conclusions?

Reviewer #2: Yes

3. Has the statistical analysis been performed appropriately and rigorously? 

Reviewer #2: Yes

4. Have the authors made all data underlying the findings in their manuscript fully available?

Reviewer #2: Yes

5. Is the manuscript presented in an intelligible fashion and written in standard English?

Reviewer #2: Yes

6. Review Comments to the Author

Reviewer #2: The authors have addressed all the concerns, the paper is improved and do not need any further adjustments from my perspective

7. PLOS authors have the option to publish the peer review history of their article (what does this mean?). If published, this will include your full peer review and any attached files.

Reviewer #2: No

---

## [Editor Report · Acceptance letter]

16 Aug 2024

PONE-D-24-17855R1 

PLOS ONE

Dear Dr. Regan, 

I'm pleased to inform you that your manuscript has been deemed suitable for publication in PLOS ONE. Congratulations! Your manuscript is now being handed over to our production team.

Kind regards, 

on behalf of

Prof. dr. Tanja Grubić Kezele 

Academic Editor

PLOS ONE